# Patterns of Medication Management and Associated Medical and Clinical Features among Home-Dwelling Older Adults: A Cross-Sectional Study in Central Portugal

**DOI:** 10.3390/ijerph20031701

**Published:** 2023-01-17

**Authors:** Maria dos Anjos Dixe, Joana Pinho, Filipa Pereira, Henk Verloo, Carla Meyer-Massetti, Sónia Gonçalves Pereira

**Affiliations:** 1Center for Innovative Care and Health Technology, Polytechnic of Leiria, 2411-901 Leiria, Portugal; 2School of Health Sciences, Polytechnic of Leiria, 2411-901 Leiria, Portugal; 3School of Health Sciences, HES-SO Valais-Wallis, 2800 Sion, Switzerland; 4Institute of Biomedical Sciences Abel Salazar, University of Porto, 4099-002 Porto, Portugal; 5Service of Old Age Psychiatry, Lausanne University Hospital, 1011 Lausanne, Switzerland; 6Clinical Pharmacology and Toxicology Unit, General Internal Medicine Clinic, University Hospital of Bern (Inselspital), 3010 Bern, Switzerland; 7Institute for Primary Health Care (BIHAM), University of Bern, 3012 Bern, Switzerland

**Keywords:** older adults, medication management, polypharmacy, multimorbidity, frailty, home-dwelling, medication-related problems

## Abstract

Ageing is frequently associated with multimorbidity and polypharmacy. The present study aimed to identify the current medication management patterns and the profiles of home-dwelling older adults and to find any association with their conditions, including frailty and cognitive impairment. Within the scope of this cross-sectional study, 112 older adults living in the community were assessed via face-to-face structured interviews. Frailty, cognitive status, medication management and clinical and sociodemographic variables were evaluated. Descriptive and inferential statistics were calculated. The mean participant age was 76.6 ± 7.1 years, 53.6% of participants were women, and 40.2% of participants lived alone. More than half were classified as having frailty (58.9%), almost one-fifth (19.6%) presented with a moderate cognitive impairment had more than one disease, and 60.7% were polymedicated. No associations were found between polymedication and medication self-management, the use of over-the-counter medications, living alone, having a poor understanding of pharmacological therapy and/or pathology, or having more than one prescriber. Self-management was associated with age, the number of medications, frailty and cognitive status. Binary logistic regressions showed that cognitive impairment had statistically significant differences with medication management, having a poor understanding of pharmacological therapy and/or pathology, having one prescriber and the use of medications not prescribed by physicians. Interventions to prevent medication-related problems in home-dwelling older adults are recommended.

## 1. Introduction

Polymedication among the older adult population is a collective health problem. In recent years, medical and pharmaceutical advances, combined with the implementation of evidence-based public health policies, have enabled clinicians to provide better healthcare and increase life expectancy. According to Portugal’s National Statistics Institute, in 2021, the country’s 65-year-olds could expect to live a further 19.35 years. According to the United Nations, in 2050, older adults (≥65 years) will make up to one-sixth of the world population [1,2].

Ageing is associated with progressive overall functional decline, making older populations more susceptible to disease. Multimorbidity—defined as the co-occurrence of two or more chronic diseases in a single individual—is also frequent in older adults [3]. Their most prevalent chronic diseases are cardiovascular diseases (70% have hypertension) [4], cancer [5], type 2 diabetes [6] and chronic respiratory diseases [3]. These diseases determine their need for diverse medication therapies, often resulting in polypharmacy [7,8], defined as the concomitant use of five or more drugs [9]. Despite innumerable undisputed benefits, pharmacotherapy remains associated with the risk of medication-related problems (MRPs), defined as unwanted effects that actually or potentially interfere with health outcomes. Recent studies have reported differing levels, but the overall median prevalence of MRPs has been found to be 70.8% (IQR = 61.0–80.2), with a range of 16.0% to 88.7% [10]. Due to the high risk of adverse drug events, prescribing appropriate medications to older adults with multimorbidity and polypharmacy can be challenging [7,8,9]. Several studies have reported that half of patients with multimorbidity do not properly adhere to their medication prescriptions [7].

In Portugal, a cross-sectional epidemiological study based on data from the first National Health Examination Survey concluded that 38.5% of the population had multimorbidity, more frequently women, with percentages increasing with age for both sexes [11]. In another Portuguese study, potentially inappropriate medication use was reported in up to 39% outpatients [12]. In addition to the medication regimen itself, non-adherence is associated with poor clinical outcomes, increases in avoidable hospital and nursing home admissions and greater healthcare costs [13]. Cognitive impairment—which can sometimes be partially worsened by specific medications, such as anticholinergics—also represents a substantial risk factor for MRPs and unplanned hospitalisation. Although often appropriate, polypharmacy is nevertheless associated with non-adherence among home-dwelling older adults, with treatment complexities, such as the number of pills per intake and the different modes of administration, being described as contributors to non-adherence [14,15,16]. One Portuguese cross-sectional study concluded that almost half of home-dwelling older adults failed to fully adhere to their medical prescriptions [15]. This was directly associated with forgetfulness, difficulties in managing medications, price, concerns about side effects, doubts and a lack of trust in some medications [15]. Polypharmacy contributes to this problem, with the number of medications being the stand-out risk factor [17], correlated with a decrease in physical function, an increase in adverse drug events, early mortality, falls and hospital length of stay [14,18,19,20].

Polypharmacy and its associated unfavourable outcomes may also play an important role in frailty syndrome [10]. Frailty is a geriatric syndrome of multidimensional origin described as a state of greater vulnerability [21,22,23]. Its prevalence among home-dwelling older adults varies from 4% to 59%, with an overall prevalence in the community estimated at near 10% [18,24]. It is commonly described and interpreted using the Tilburg Frailty Index (TFI), which analyses the physical, psychological and social dimensions of frailty. TFI is considered one of the most robust instruments for assessing frailty, especially in primary healthcare [25]. The early detection of frailty and subsequent earlier interventions carried out to prevent its progression are important in order to enable older adults to remain living at home [17]. Moreover, frailty among home-dwelling older adults is associated with adverse outcomes, such as disability, falls, a lower quality of life, hospitalisation, institutionalisation and multimorbidity [7,21], all of which are specific risk factors for MRPs [10].

Despite ageing and its associated multimorbidity, polypharmacy and frailty, older adults can remain at home if their health conditions are well managed, including using an adequate therapeutic regimen based on an interprofessional collaborative approach. This is beneficial not only for the older adult’s overall health status but also for the healthcare system’s finances [10,21].

Patterns of medication management are under-reported and under-investigated. Developing a greater understanding of current medication practices can help healthcare systems to propose more efficient recommendations in order to prevent MRPs. Given the above, the present study thus aimed to identify the current medication management patterns and the profiles of home-dwelling older adults by considering their medical conditions, frailty and cognitive status. Additionally, this study explored the associations between health conditions, including frailty and cognitive impairment, and medication management, and the predictors between sociodemographic characteristics and MRP patterns.

## 2. Materials and Methods

### 2.1. Study Design

This cross-sectional, observational study was conducted following the recommendations of the Strengthening the Reporting of Observational Studies in Epidemiology (STROBE) statement [26].

### 2.2. Participants

Data were collected from home-dwelling older adults (≥65 years) visiting a single primary healthcare centre in the Leiria region in Portugal. They were recruited using convenience sampling on the day that they visited for a consultation. A 15 to 20 min face-to-face structured interview was conducted after assessing the older adult’s cognitive status using the Six-Item Cognitive Impairment Test (6-CIT) [27]. If they scored ≥10, they were interviewed directly; if they scored <10, the structured interview was conducted with their caregivers, who accompanied them in the consultation.

### 2.3. Data Collection

Data were collected two days per week in March and April 2022, with a mean of ten participants per day. The evaluation instruments were applied by a research assistant trained in communication with older adults.

The structured interview included the following assessments:

(a) Sociodemographic variables: age, sex, marital status, household status, education and income.

Clinical variables: the number of medical conditions (using the International Classification of Diseases—version 10 (ICD–10)), the number of hospitalisations/emergency department admissions and other medical consultations in the last 12 months. Frailty was assessed using the psychometrically validated Portuguese version of the Tilburg Frailty Index (TFI-P) [28] subscale B, which includes 8 questions assessing physical frailty, 4 questions assessing psychological frailty and 3 questions assessing social frailty [28,29]. The older adults who scored ≥5 on the TFI-P subscale B were classified as having frailty [28]. Cognitive status was assessed using the 6CIT-P, with a cut-off score at 5 and higher scores indicating a higher cognitive impairment [30].

(b) Medication management variables: medication name/classification (according to the Anatomical Therapeutic Chemical Classification) [30], polypharmacy (the number of medications ≥ 5, including those to be taken only in an emergency, SOS), the identification of the person managing the older adult patients’ medication, the use of medications not prescribed by physicians, the understanding of one’s own medication prescription and/or disease (cognitive impairments and/or communication problems, e.g., foreign language and hearing impairments) and the number of medication prescribers.

### 2.4. Data Analysis

Descriptive univariate and bivariate statistical analyses were computed using SPSS (version 28.0) software (IBM, Armonk, NY, USA). Measures of central tendency and dispersion were used for quantitative variables, and either frequencies or percentages were used for qualitative variables. Inferential statistics were used to study the relationships between variables. Since the data did not present a normal distribution according to the Kolmogorov–Smirnov test, Mann–Whitney U tests were used instead. Next, binary logistic regressions were performed to determine which sociodemographic and clinical conditions were predictors of different medication management patterns, with odds ratios (ORs) calculated with a 95% confidence interval. A *p* value of less than 0.05 was considered statistically significant.

### 2.5. Ethical Considerations

This study was conducted in accordance with the Declaration of Helsinki, and the study protocol was approved by the Ethics Committee of the Regional Health Administration of Leiria, Portugal (80-2021). Written informed consent was obtained from all participants. Data confidentiality was ensured by anonymous coding and storage in restricted-access files.

## 3. Results

### 3.1. Sociodemographic Characteristics

A total of 112 older adults participated in our study. After applying the 6-CIT questionnaire, three caregivers answered in the interviews instead of the older adult. The mean age of the 112 participants was 76.6 ± 7.1 years (median = 76), with 53.6% of the participants being women and 46.4% being men. Most of the participants were born in Portugal (93.8%), were married (59.8%), and lived with a partner or spouse (61.4%), and about three-quarters (70.5%) only had a basic level of education of ≤4 years (Table 1). More than half (58.1%) had an income lower than the Portuguese national minimum wage, defined as being below the poverty line [31].

### 3.2. Medication Management Patterns, Profiles and Clinical Conditions

Most of the home-dwelling older adult participants (*n* = 94; 84.3%) managed their own medication, with 14 (12.2%) needing support from their first- or second-degree relatives and 4 (3.5%) requiring professional support. Sixty-eight (60.7%) were classified as being polymedicated, with the majority of medications being prescribed by their family physician. When asked about over-the-counter medication, 32 (28.6%) participants declared that they take it, mainly analgesics, non-steroidal anti-inflammatory drugs, vitamins and magnesium and calcium supplements, while 80 (71.4%) participants declared that they did not. Most older adults (104; 92.9%) had a personal community pharmacist, and 8 (7.1%) had multiple pharmacists.

Regarding the participants’ clinical conditions, the majority (*n* = 102; 91.1%) presented with multimorbidity, with an average of 3.5 ± 1.5 chronic diseases. In the 12 months prior, 87 (77.6%) participants had an average of 2.87 ± 3.4 medical consultations, mainly with family practitioners and nurses; 25 (22.3%) participants were admitted to the emergency department.

Following evaluation using the 6CIT-P, most participants showed no cognitive impairments (*n* = 90; 80.4%), whereas 22 (19.6%) were classified as having a moderate cognitive impairment. Based on the TFI-P subscale B 5-point cut-off, 58.9% of the older adults were considered to have frailty, with higher scores for social frailty (Table 2).

### 3.3. Associations between Patient-Level Medication Management Patterns and Their Sociodemographic and Clinical Characteristics

In the present study, the patients with multimorbidity (*n* = 100) took more (5.8 ± 3.2; median = 6) medications (U = 980.000; Z = −2.804; *p* = 0.005) than the patients (4.1 ± 2; median 6) not classified as having multimorbidity (*n* = 12), and those who took more medications were more often responsible for managing their own (*n* = 94; 5.7 ± 3.2 median = 6) medications (U = 30.000; Z = −3.640; *p* < 0.001) than those who need help from informal caregivers (*n* = 16; 5.5 ± 4; median = 5).

No associations were observed between being a polymedicated patient and medication self-management (*p* = 0.343), using over-the-counter medication (*p* = 0.081), living alone (*p* = 0.625), not fully understanding one’s own prescribed medication and/or disease (*p* = 0.023) or having more than one prescriber (*p* = 0.893).

The older adults with only one prescriber had a statistically significant (U = 748.00; Z= −3.196; *p* =0.001) higher 6CIT-P score (9.0 ± 6.1; median = 8) than those with more than one prescriber (5.5 ± 5.2; median = 4). The patients who took over-the-counter medication had statistically significant (*p* = 0.049) lower 6CIT-P scores (5.9 ± 3.2; median = 6) than those who did not (9.0 ± 6.7; median = 8).

The medication management variables (who oversees the medication preparation, the number of over-the-counter medications, the number of prescribed medications taken per day and the understanding of one’s own medication prescription and/or disease) were not statistically associated with sex or household status. However, the self-management of medication showed a statistically significant correlation with age (*p* < 0.001), cognitive status (*p* < 0.001) and frailty (*p* < 0.001), with those responsible for their own medication management being younger (75.6 ± 5.6; median = 75.5) than those who were not (82.5 ± 11.3; median = 85), presenting with a better cognitive status score (6.7 ± 4.4; median = 8 vs. 18.0 ± 7.3; median = 20) and a lower frailty index score (4.9 ± 3.0; median = 5 vs. 8.5 ± 2.5; median = 9) (Table 3).

In order to verify which factors are the predictors of medication management patterns, a binary logistic regression was performed (Table 4). Only the cognitive impairment variable presented statistically significant correlation probability values with the independent variables, being the highest among older adults who had medication management support (OR = 0.197; 95% CI: 0.046–0.853), who reported problems understanding their medication prescription and/or their disease (OR = 0.169; 95% CI: 0.051–0.566) and who did not take over-the-counter medications (OR = 0.280; 95% CI: 0.097–0.803).

## 4. Discussion

The present study explored the patterns of medication management among polymedicated and non-polymedicated home-dwelling older adults in central Portugal. Because some of the medication management patterns identified (e.g., polypharmacy, multiple prescribers and a poor understanding of prescribed medications) could lead to MRPs, our results highlight the potential factors influencing medication safety in this population. This study also sought associations between medication management patterns and sociodemographic characteristics, clinical or medical conditions, frailty and cognitive status. More than three-quarters of the participants had a low level of education and lived in poor socioeconomic conditions. Previous studies have considered these influences as potential risk factors for non-adherence to or the cessation of prescribed medications [32,33,34,35,36]. Drug interactions, overuse and the wrong drug choice can also contribute to this situation [37]. The medication management patterns of one-third of the sample potentially increased their risks of an MRP (the use of potentially inappropriate medicines, drug–drug interactions and hospitalisation) due to self-treatment with unprescribed over-the-counter medication, as previously reported by Jin [38]. Most of our home-dwelling older adults were polymedicated but, nevertheless, managed their medications without regular support from informal or professional caregivers, although this increases the risks of errors, compliance issues and suboptimal medication management [39]. Our findings show that one-fifth of the studied sample presented with a moderate cognitive impairment; this is often associated with an increased risk of non-adherence, which can lead to poor therapeutic outcomes, greater risks of MRPs and poor medication management, as reported in the systematic review by Kröger [40]. In the same review, the authors concluded that, in older adults, cognitive disorders contribute to worse adherence to medication prescriptions. However, our findings show that the older adults with cognitive impairments received the most robust support from formal and informal caregivers, thus potentially protecting them more from inappropriate medication management (OR = 0.197), the use of over-the-counter medication (OR = 0.280) and issues resulting from a limited understanding of their medication prescription and disease (OR = 0.169). These results are in line with the findings of other authors [41] and the meta-analysis by DiMatteo [42]. The latter authors reported that social support measures, such as structured professional networks, were positively linked to medication adherence among home-dwelling older adults regardless of whether they had cognitive disorders. Our findings found no medication management issues associated with sex or household status. Contrarily, medication self-management was significantly related to age (younger participants were more autonomous), better frailty scores and good cognitive status, all in line with previous studies [43].

Almost all of the participants presented with multiple chronic conditions (91.1%), leading to polypharmacy in more than half of them. More than half (58.9%) were also shown to have frailty based on their elevated scores on the highly sensitive TFI scale’s social frailty questions. Considering the sample’s combination of multimorbidity, polypharmacy and high prevalence of frailty, our participants were exposed to a high risk of MRPs. This finding is corroborated in other studies [44,45] reporting that age-related factors, such as multimorbidity, chronic pain, polypharmacy and frailty, should be considered when prescribing medication. Taken together, these findings lead us to recommend the implementation of strategies to reduce modifiable risk factors—such as pre-frailty and moderate frailty—and to optimise medication adherence, reduce medication over-use and avoid medication under-use, all of which could be effective measures for decreasing the risk of MRPs and helping older adults to remain in their homes. Several authors [43,46,47,48], from different countries, have stated that interventions for reducing potentially inappropriate medicines and promoting medicine optimisation might improve the suitability of the medication of home-dwelling older adults with frailty. However, we found no significant associations between polypharmacy and medication self-management, using over-the-counter medications, living alone, not fully understanding one’s own pharmacological therapy and/or pathology, or having more than one prescriber. Our relatively small sample probably prohibited us from identifying statistically significant differences in these characteristics.

### Study Strengths and Limitations

Our study strengths include the rigorous application of our inclusion and exclusion criteria for recruiting participants for a face-to-face interview. All data were collected in structured interviews to ensure that they were complete.

Regarding the limitations, our relatively small sample was not based on a calculation of necessary statistical power. Another concern is that the study also used some self-reported data and did not investigate medication adherence directly via pill counts and evidence of over-the-counter medication. The cross-sectional research design limited any follow-up on the evolution of medication management patterns and the associated risks of MRPs. Some data on historical events were collected, and memory problems could have compromised data accuracy. Finally, the face-to-face structured interviews could have resulted in some participants providing socially desirable answers.

## 5. Conclusions

Home-dwelling older adults are exposed to higher risks of MRPs due to having frailty and multimorbidity, associated with polypharmacy and exacerbated by cognitive impairment. However, in the present study, cognitive impairment was found to be a protective factor against MRPs, as the older adults with cognitive impairments received more robust medication management support from their informal and professional caregivers. A multidisciplinary, collaborative approach that supports and promotes the management of home-dwelling older adults’ medications in order to prevent MRPs and subsequent institutionalisation is important. This could be especially relevant among home-dwelling older adults presenting with deficits in health and medication literacy and among those using over-the-counter medications. Indeed, and based on our findings, instead of waiting for the cognitive capacities of older adults to decline before providing them with medication management support, efforts should be made to do so pre-emptively before they reach more advanced states of frailty. This approach could translate into important financial savings for older adults and their families, as well as for national healthcare systems and society overall, in addition to ensuring a better quality of life for all.

## Figures and Tables

**Table 1 ijerph-20-01701-t001:** Participants’ sociodemographic characteristics (*n* = 112).

Variables		*n*	%
Marital Status	Married	67	59.8
	Unmarried	7	6.3
	Separated/divorced	11	9.8
	Widow/widower	27	24.1
Household status (Who do you live with?)	Alone	29	25.5
	Spouse/partner	70	61.4
	Children	12	10.5
	Others	3	2.6
Educational level	None or 1st cycle (1st to 4th school years)	79	70.5
	2nd cycle (5th and 6th school years)	7	6.2
	3rd cycle (7th to 9th school years)	9	8.0
	Secondary education (8th to 12th school years)	12	10.7
	Higher education (professional or university)	5	4.5

**Table 2 ijerph-20-01701-t002:** Participants’ medication management, medical conditions and TFI-P subscale B results determined in the structured interviews (*n* = 112).

	Variables		*n*	%	Mean/SD
How many prescribed medications do you currently take per day?	≤4 medications		44	39.3	
	≥5 medications		68	60.7	
Number of hospitalisations/emergency department admissions in last 12 months	0		87	77.7	
	1		21	18.8	
	2		2	1.8	
	4		1	0.9	
	5		1	0.9	
Reasons for the above admissions	Pain (back; legs; arms; others)		5	22.8	
	Falls		4	18.2	
	Urinary tract infection		4	18.2	
	Fracture (leg; wrist; others)		2	9.0	
	Heart problems		4	18.2	
	Stroke		1	4.6	
	Psychiatric disorders		2	9.0	
Medical consultations in the last 12 months	Yes		87	77.6	
	No		25	22.4	
Which medical specialists were consulted?	Family practitioner		102	49.3	
	Nurse		63	30.4	
	Cardiologist		32	15.5	
	Diabetologist		4	1.9	
	Geriatrician		0	0.0	
	Psychiatrist		6	2.9	
Who prescribed your current medications?	Family practitioner		110	76.9	
	Cardiologist		16	11.2	
	Diabetologist		1	0.7	
	Geriatrician		0	0.0	
	Psychiatrist		5	3.5	
	Others		11	7.7	
Medical conditions (ICD–10)	Diseases of the circulatory system		108	27.0	
	Endocrine, nutritional and metabolic diseases		65	16.2	
	Diseases of the musculoskeletal system and connective tissue		49	12.2	
	Diseases of the genitourinary system		48	12.0	
	Diseases of the blood and blood-forming organs and certain disorders involving the immune mechanism		7	1.8	
	Mental and behavioural disorders		45	11.2	
	Neoplasms		30	7.5	
	Diseases of the ear and mastoid process		7	1.8	
	Diseases of the eye and adnexa		7	1.8	
	Diseases of the respiratory system		13	3.3	
	Diseases of the digestive system		21	5.2	
Understanding of medication prescribed and/or the disease	No		70	62.5	
	Yes		42	37.5	
TFI-P physical frailty (0–8)					3.04 ± 1.98
TFI-P psychological frailty (0–4)					1.55 ± 1.16
TFI-P social frailty (0–3)					0.80 ± 0.91
TFI-P total frailty score (0–15)					5.40 ± 3.18

Only 4 participants remembered the date.

**Table 3 ijerph-20-01701-t003:** Associations between medication management patterns and sociodemographic and clinical characteristics of the studied home-dwelling older adults (*n* = 112).

Older Adults’ Variables		Cognitive Impairment *				Frailty *				Age *				Number of Medical Conditions **				Income **	Sex **
		P_25_	P_50_	P_75_	*p* **	P_25_	P_50_	P_75_	*p*	P_25_	P_50_	P_75_	*p*	P_25_	P_50_	P_75_	*p*	*p*	*p*
In charge of medication management	Own	4	8	10	**<0.001**	7	9	11	**<0.001**	79.2	85	89	**<0.001**	1	2	3	**<0.001**	0.089	0.296
	Other	10	20	22		3	5	7		72	75.5	79		2.7	4	5			
Use of over-the-counter medications without prescription	No	4	8	12	**0.049**	3	5	7	0.892	72	76	81.7	0.897	2	3	4	**0.036**	0.973	0.881
	Yes	4	6	8		3.2	6	7		73	76.5	80		3	4	5			
Number of prescribed medications currently taken per day	0–4	4	8	14	0.148	2.2	6	7.7	0.926	73.2	78	82.5	0.132	2	3	4	**0.005**	0.318	1.000
	>5	4	8	10		3	5	7		71.2	75	81		3	4	5			
Understanding of medication prescribed and/or the disease	No	2	4	8	**<0.001**	2	4	7	**<0.001**	71	74	78	**<0.001**	3	4	5	0.136	**0.036**	0.696
	Yes	10	12	18.5		4.7	7	8.2		75	81.5	8.2		2	3	5			

Note: * Mann–Whitney test; ** chi-squared test; statistically significant results are highlighted in bold.

**Table 4 ijerph-20-01701-t004:** Binary logistic regressions of medication management patterns and sociodemographic and clinical characteristics (*n* = 112).

Dependent Variables *	Sex (M/W)			Age			Cognitive Impairment (No/Yes)			Frailty (No/Yes)			Number of Medical Conditions (No/Yes)			Income		
	M/W	OR (95% CI) †	*p*-Value	65–74/≥75	OR (95% CI) †	*p*-Value	No/Yes	OR (95% CI) †	*p*-Value	No/ Yes	OR (95% CI) †	*p*-Value	1/>1	OR (95% CI) †	*p*-Value	≤EUR 705/>EUR 705	OR (95% CI) †	*p*-Value
Who oversees medication preparation? (1)																		
Older adult (1)	47/49	0.972 (0.242–3.913)	0.969	42/54	0.161 (0.018–1.434)	0.102	67/29		0.030	46/50	1920 (1585–2326)	0.997	5/91	3.72 (0.641–21.69)	0.143	29/67	0.460 (0.046–4.64)	0.511
Other (0)	5/11			1/15			3/13	0.197 (0.046–0.853)		0/16			5/11			1/15		
Do you take over-the-counter medications without a prescription? (1)																		
No (0)	38/42		0.923	32/48		0.171	45/35	0.280 (0.097–0.803)	0.018	35/45		0.157	10/70		NA **	22/58		0.815
Yes (1)	14/18	1.04 (0.406–2.70)		11/21	1.92 (0.754–4.93)		25/7			11/21	2.04 (0.758–5.53)		0/32	NA **		8/21	1.13 (0.390–3.30)	
N medications currently taken per day (1)																		
0–4 (0)	20/24		0.887	14/30		0.452	24/20		0.365	19/25		0.215	6/38		0.264	9/35		0.232
>5 (1)	32/36	0.941 (0.404–2.19)		29/39	0.718 (0.302–1.70)		46/22	0.666 (0.276–1.606)		27/41	1.77 (0.718–4.36)		4/64	2.21 (0.548–8.97)		21/47	0.546 (0.202–1.47)	
Understanding of medication prescribed and/or the disease (1)																		
No (0)	34/36		0.369	35/35		0.830	70/0	0.169 (0.051–0.566)	0.004	36/34		0.341	3/67		0.13	46/24		0.748
Yes (1)	18/24	0.642 (0.245–1.68)		8/34	1.10 (0.436–2.81)		0/42			10/32	1.63 (0.595–4.48)		7/35	2.55 (0.271–23.91)		36/6	0.845 (0.302–2.36)	

† OR, odds ratio; CI, confidence interval; * dummy variable (0 = no; 1 = yes); ** not applicable.

## Data Availability

The datasets used and analysed during the current study are available from the corresponding author on reasonable request. Most data generated or analysed during this study are included in this published article.

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
