# Peer review of "Patterns of Medication Management and Associated Medical and Clinical Features among Home-Dwelling Older Adults: A Cross-Sectional Study in Central Portugal"

_ijerph, 2023, doi:10.3390/ijerph20031701_

Round 1

Reviewer 1 Report

2.2, 3rd sentence - remove one occurrence of the word structured to make the sentence make more sense.

2.4 - It is not clear what the outcome(s) of the logistic regression are.

Table 2 - where you say 4 pills, 5 pills - do you mean medications?  People could have multiple pills of the same medication each day.

P6, 1st sentence - it is unclear which two groups are being compared with these Mann Whitney U tests.  Also some descriptive statistics for the two groups would be useful to see the magnitude of the differences.

P6, 2nd, 3rd and 4th paragraphs - some descriptive statistics would be useful to show the reader the magnitude of the differences.  For those where you provide some descriptive statistics and perform Mann Whitney U test the appropriate descriptive statistics are median (IQR) for the two groups (not mean (SD)).

P6, 4th paragraph - where you say correlated, you probably mean associated, please change the wording.  You should only use correlated when you are referring to the results of correlation.

P7 - change correlated to associated

Table 4 - oversees medication preparation older adult/ frailty 95% CI cannot be 0.000, 0.000

P9 Discussion - change correlations to associations or relationships.

Author Response

Please see attached document with response

Reviewer 2 Report

This manuscript addresses medical management among community dwelling elderly patients in a community in Portugal. Overall, the manuscript is well-written with sound methodology.

There are a few typographical and syntax. Examples are

Abstract - "Line 6"; "were evaluuted" - Change to "evaluated"

Results - (b) - "declared not to take over-the-counter medication": This could be re-phrased to give better clarity.

Author Response

Please see attached one document with author´s  answers 
